# Defective RNA Particles of Plant Viruses—Origin, Structure and Role in Pathogenesis

**DOI:** 10.3390/v14122814

**Published:** 2022-12-16

**Authors:** Daria Budzyńska, Mark P. Zwart, Beata Hasiów-Jaroszewska

**Affiliations:** 1Department of Virology and Bacteriology, Institute of Plant Protection-National Research Institute, Wl Wegorka 20, 60-318 Poznan, Poland; 2Department of Microbial Ecology, Netherlands Institute of Ecology (NIOO-KNAW), Droevendaalsesteeg 10, 6708 PB Wageningen, The Netherlands

**Keywords:** plant viruses, subviral particles, defective RNA particles, DI RNAs

## Abstract

The genomes of RNA viruses may be monopartite or multipartite, and sub-genomic particles such as defective RNAs (D RNAs) or satellite RNAs (satRNAs) can be associated with some of them. D RNAs are small, deletion mutants of a virus that have lost essential functions for independent replication, encapsidation and/or movement. D RNAs are common elements associated with human and animal viruses, and they have been described for numerous plant viruses so far. Over 30 years of studies on D RNAs allow for some general conclusions to be drawn. First, the essential condition for D RNA formation is prolonged passaging of the virus at a high cellular multiplicity of infection (MOI) in one host. Second, recombination plays crucial roles in D RNA formation. Moreover, during virus propagation, D RNAs evolve, and the composition of the particle depends on, e.g., host plant, virus isolate or number of passages. Defective RNAs are often engaged in transient interactions with full-length viruses—they can modulate accumulation, infection dynamics and virulence, and are widely used, i.e., as a tool for research on *cis*-acting elements crucial for viral replication. Nevertheless, many questions regarding the generation and role of D RNAs in pathogenesis remain open. In this review, we summarise the knowledge about D RNAs of plant viruses obtained so far.

## 1. Introduction

Viruses constitute the most abundant biological entities in the biosphere. They exhibit highly heterogeneous genome structures, sizes and replication strategies.

High diversity of viral nucleotide sequences is a characteristic feature of many viruses, in particular the RNA viruses. RNA viruses typically have mutation rates between 10^−6^ and 10^−4^ substitutions per nucleotide per cell, whereas these rates range from 10^−8^ to 10^−6^ in DNA viruses [1]. Plant viruses are dominated by RNA viruses [2], which tend to have a great potential for genetic variation, rapid evolution and adaptation. Different mechanisms that generate diversity in viral genomes have been investigated: mutation, recombination and reassortment [3]. These mechanisms can all contribute to the formation and evolution of defective RNAs (D RNAs). D RNAs are non-infectious virus particles harbouring defective RNA, derived from full-length viral genomes—termed “helper virus” (HV)—that cannot replicate autonomously. D RNAs are *trans*-replicated by the HV, and only those that interfere with HV accumulation should be referred to as defective interfering particles or RNAs (DIPs or DI RNAs). DI RNAs are, therefore, a subclass of D RNAs, and for accuracy, we will distinguish between these two classes throughout this review.

The first information about DI RNAs originated in 1947 when Preben von Magnus described non-infectious, incomplete forms of influenza A virus (IAV) [4]. This form of viral genome arose spontaneously during passaging of infectious virus at a high multiplicity of infection (MOI) [5]. In subsequent years, the presence of D RNAs was confirmed for many other animal RNA viruses, e.g., vesicular stomatitis virus (VSV) [6,7] or Sendai virus (SeV) [8]; however, the term DI RNA was used for the first time by Huang and Baltimore in 1970 [9]. Their research first confirmed previous inklings that DI RNAs were incomplete viral genomes that interfered with the replication of the wild type virus (i.e., full-length HVs), and that their formation is a crucial facet of the virus evolution [9]. A wide range of observations in many virus–host systems have led to the conclusion that D RNAs are pervasive within animal and human RNA viruses, which has been confirmed for representatives of numerous families in vitro [10]. It has been recently confirmed that D RNAs of human and animal viruses can be detected in vivo in patients infected with, e.g., hepatitis C virus (HCV) [11], or in birds infected with West Nile virus (WNV) [12]. Extensive studies on DI RNAs of animal and human viruses showed their participation in modulating virus replication, influencing disease outcome, triggering immune responses (i.e., through inducing the expression of interferon (IFN), tumour necrosis factor (TNF) or interleukin (IL)) and promoting virus persistence following infections. Therefore, these results make them possible candidates for vaccine adjuvants and antivirals [10,13].

Despite the fact that plant viruses were discovered first (at the end of the 19th century), the earliest reports of their D RNAs appeared in 1983; D RNA formation was suspected for potato yellow dwarf virus (PYDV) [14] and tomato spotted wilt virus (TSWV) [15]. In both cases, identification of the D RNAs was performed based on their buoyant density or electron microscopy, but the presence of the D RNAs was not confirmed using molecular methods. More advanced research was performed by Hillman et al. (1987) [16], who presented both physical and biological properties of D RNAs associated with cherry strain of tomato bushy stunt virus (TBSV). Afterwards, the presence of D RNAs was confirmed for cymbidium ringspot virus (CymRSV) (1989) [17], turnip crinkle virus (TCV) (1989) [18], cucumber necrosis virus (CNV) (1991) [19], cucumber mosaic virus (CMV) (1995) [20], and several other plant viruses. To date, the existence of D RNAs/DI RNAs has been confirmed for at least 15 plant viruses, all of which occur de novo upon single infection or serial passaging (Table 1). For many of these viruses, their artificial D RNAs were constructed to investigate mechanisms of viral infection (e.g., as done for tobacco mosaic virus, TMV [21]). Moreover, defective and defective interfering viruses have also been found for many DNA viruses, e.g., tomato yellow leaf curl virus (TYLCV) or maize streak virus (MSV) [22,23,24].

The D RNAs associated with plant viruses are highly variable even within species, including their nucleotide structure, host dependence or effect on HVs. Here, we summarise the current knowledge about D/DI RNAs associated with plant RNA viruses.

## 2. Generation of D RNAs and Mechanism of Induction

Two different stages can be distinguished during D RNA generation: (i) particle formation during replication process through recombination events, and (ii) selection of newly formed D RNAs, resulting in their accumulation [54].

D RNAs of plant viruses typically arise when a virus is serially passaged in conditions of high cellular MOI. Hypotheses concerning D RNA formation were widely investigated for many years, and a number of conclusions can be drawn. Firstly, although a positive effect of high MOI on D RNA formation was confirmed, for some virus species, high MOI is indispensable (e.g., TBSV) [25]**,** whereas for others, it is only considered as an advantage [55]. High MOI favours coinfection of plant cells both with D RNAs and HV [56], allowing the D RNAs to be maintained and selected. D RNA formation is contingent on virus species (or even viral isolates), and for several viruses, low MOI at the start of serial passaging does not impede D RNA formation (e.g., CNV) [19]. By contrast, this situation seems to be uncommon, and it may explain the paucity of D RNA formation in natural environments. It is generally believed that D RNAs do not accumulate to easily detectable levels during natural virus infections. Low MOIs in plants in vivo can prevent the formation, accumulation or spread of D RNAs in nature [57]. Secondly, a single virus isolate can generate more than one species of D RNA de novo [25,47,50,58]. During the continuous passages, the D RNA population can evolve from a heterogeneous to homogenous one [27]**,** and the different electrophoretic patterns of D RNAs can be observed during different stages of prolonged passaging [28]. It is not only possible that new D RNAs are continually generated from HVs in a process with a degree of repeatability; theoretical work also suggests that D RNAs evolve continuously [59]. The number of passages necessary for D RNA detection can vary for different virus species and isolates [25,58]. For instance, the occurrence of TBSV DI RNAs during prolonged passaging of HV in tobacco was observed even as early as after the third passage for some lines, and as late as after the 11th passage [25]. As genetic drift affects the maintenance of beneficial mutations in a population [60], bottlenecks that occur during within-host, or between-host transmission could purge DI RNAs and have a strong stochastic effect on their accumulation over time. By contrast, often the structure of D RNAs that arise independently from the same virus species, or even different species of the same genera, is conservative to a certain extent, and newly formed molecules are composed of the same regions of viral genomic sequences (e.g., *Tombusvirus* genus) [61]. Finally, the mechanisms of D RNA generation and maintenance appear to be host-specific, as has been confirmed for many D RNA species. Passaging of TBSV in *Nicotiana benthamana* resulted in DI RNA generation and accumulation, whereas prolonged passages of the same virus isolate in pepper did not result in generation and accumulation of DI RNAs, even in the case of inoculated leaves [56]. This situation can be explained by interaction between viral elements and host-specific determinants, or changes in the demography of virus populations (i.e., different MOI).

To date, the propensity for D RNA induction has been confirmed for plant viruses from different genera, i.e., *Tombusvirus*, *Orthospovirus*, *Closterovirus*, *Tobravirus*, *Potexvirus*, *Bromovirus*, *Cucumovirus*, *Crinvirus*, *Comovirus*, *Nepovirus* or *Pomovirus* (Table 1). Hopping or template switching of viral RdRp (also called copy-choice mechanism) is considered to be the most probable mechanism of D RNA formation [62]. D RNAs are mainly derived from the genome of the HV; premature dissociation of viral RNA polymerase and nascent strand from the RNA template is followed by reinitiation of replication after binding to the same or corresponding template at a different site, resulting in newly synthesized, incomplete strands [62,63]. Hence, D RNAs are derived from the HV genomic RNAs through a copy choice mechanism resulting in sequence deletion(s).

Analysis of regions near the apparent junction sites in D RNA sequences revealed that homologous recombination plays a major role in producing D RNAs. These recombination events are, therefore, not entirely stochastic events, but rather they tend to occur between similar sequences (or stretches) with weak secondary structure [28]. Recombination-prone sites (i.e., “hotspots”) are thought to be surrounded by secondary structures with negative free energy, which are difficult to process for the polymerase and promote dissociation of RNA polymerase-nascent strand complex [28]. This hypothesis seems to be adequate for many D RNA species (e.g., CymRSV, TBRV) [28,49,50,51]. Regions near the junction sites that can potentially destabilise the polymerase complex were described for TBSV. Knorr et al. [25] linked D RNA formation in TBSV with the presence of the hexanucleotide motif 5′-APuAGAA-3′, several “strong stop” signals and the presence of inverted repeats. Hernandez et al. (1996) [37] noted that junction sites in RNA2 of TRV DI RNAs are flanked by short nucleotide repeats or sequences resembling the 5′ end of genomic and sub-genomic RNAs of TRV. The sequence motifs (i.e., AGAAAAG in *RdRp* coding region), together with complementary inverted repeats (i.e., CUUUUCU in 5′ UTR sequence) were also found near the genomic sequences of TBRV isolates [50,64].

Reinitiation of the replication process requires the presence of replicated sequence initiation signals. It was confirmed that, in the case of CymRSV DI RNAs, many junction sites start with G followed by A, which is part of the motif recognized in all replicating molecules of the virus [65]. For the same virus species, it was suggested that DI RNA monomers arose from head-to-tail dimers [66,67]. Co-inoculation of tobacco plants with transcripts of HV and short DI RNAs resulted in accumulation of de novo generated DI RNA. Moreover, those results suggest that DI RNA dimers are preferred over monomers in movement from cell to cell [67].

A replicase-driven template switching mechanism was insightfully investigated for DI RNAs derived from PMTV [52]. The mechanism of DI RNA biogenesis proposed by the authors assumes that during HV replication, base pairing occurs between the TGB1 and 8K ORF coding regions on the minus strand. As a consequence, a stem-loop structure is formed, and the DI RNAs are generated during the positive strand RNA synthesis. A minimum free energy of the structure conditioning the occurrence of the process is −13.7 kcal/mol. The high predicted stability of this structure may explain the high repeatability of the junction site’s location during the de novo generation of PMTV DI RNAs. Moreover, the other secondary RNA structures in close proximity to the 8K coding region and within the DI RNAs may be required for DI RNA biogenesis [52]. Quito-Avila et al. [68] indicated the that 5′ to 3′ pairing can act as a mechanism to generate new variants of raspberry bushy dwarf virus (RBDV), leading to formation of new, large-scale genomic rearrangements. Therefore, this mechanism may also contribute to D/DI RNA formation.

It has been shown that viral proteins can be involved in D RNA formation. The importance of the viral helicase (1a) and polymerase (2a) in generation of D RNAs was confirmed for BMV. Introduction of mutations within the corresponding coding regions resulted in changes in the fidelity of the replication and the position of recombination sites in comparison with the wild type virus [63,69].

## 3. Structure of D RNAs and Their Population

D RNAs are rearranged versions of viruses that have lost functions essential for autonomous replication. Drastic truncations and modifications of the viral genome during D RNA formation result in particles containing mostly parts of noncoding regions of the ancestral HV and the remnants of disrupted open reading frame(s) [44,63,70]. Two different types of D RNA can be distinguished: (i) the first one includes D RNAs constituting different fragments of the HV genome, emerging by multiple deletion in genomic RNA of the virus [25,61,66]; (ii) D RNAs of the second type arise as a result of large, single deletion in the genome of the HV [50,71] (Figure 1). In both cases, D RNAs may represent both homogenous or heterogeneous subpopulations [50,65]. Moreover, D RNAs can also constitute a mosaic of fragments originating from different segments of the HV genome (e.g., TRV) [37] or even a mosaic of plant and virus sequences [16] (Figure 1).

Despite having the same overall structure, the D RNA subpopulation generated during the same experiment from a single ancestral variant can be composed of different variants carrying additional small deletions and nucleotide substitutions [25,58]. The smaller D RNA particles probably arose from larger D RNA precursors [58], and their accumulation seems to be favoured over longer ones [28]. Frequently, DI RNAs of the same virus isolate (generated by passaging an isolate in different plants) are formed from the same fragments of the HV genome, and differences in length of the D RNAs proceed from shifts of recombination sites in genomic RNA [50]. The existence of some conservatism in the composition of D RNAs derived from genomic RNA of different isolates of the same virus species was confirmed. For instance, TBRV DI RNAs observed to date are distributed across the three essential types, and two of them include DI RNAs associated with TBRV isolates originated from distant host plants, such as zucchini, tomato and black elderberry [49] (Figure 2). The evolutionary repeatability of D RNA emergence was also confirmed for D RNAs of different virus species belonging to the same genus. Comparisons of DI RNA sequences associated with representatives of *Tombusvirus* genus—CSV and TBSV—demonstrate that in both DI RNAs, similar regions of the HV are present, which are essential for effective amplification and accumulation of the DI RNAs [27]. Tombusvirus DI RNAs indicate a common structural organization and contain three conserved sequence blocks referred to as A, B and C, derived from the 5′ terminus, internal region of the replicase, and the 3′ terminus of the viral genome, respectively [58,61,72]. DI RNAs associated with CymRSV and the cherry strain of TBSV share the same basic structure, and the nucleotide identity between them ranged from 80% to 90% [65]. Frequently, the invariant segments correspond to regulatory sequences that are important or crucial for D RNAs viability.

The host effect on the population structure of D RNAs was also analysed. It was confirmed that passaging a viral population containing D RNAs through different hosts promotes changes in the nature of the dominant subpopulation of D RNAs [27].

## 4. *Cis*- and *Trans*- Regulation of DI RNA

Formation and accumulation of D RNAs is controlled both by *cis*-acting elements present in the HV and their cognate D RNAs, as well as *trans*-acting components such as non-structural proteins of the HV [30]. *Cis*-acting elements are responsible for recruitment to the site of replication and assembly of the viral replicase, constituting signal sequences necessary for movement that have been described for many plant viruses [73,74,75,76]. Therefore, numerous studies were performed to establish their placement in D RNAs and corresponding sequences in HV genomes. Research performed with infectious clones of D RNAs has enabled the identification of *cis*-acting elements required for defective particle accumulation [72,77]. Efficient replication and accumulation of tombusvirus DI RNAs are regulated by four conserved genomic RNA segments essential for replication, accumulation, competitiveness and supressing activity [27,73,77,78]. The potential *cis*-acting elements responsible for encapsidation were also described for BMV D RNAs. Using artificial D RNAs (with deletions of the same size as the naturally occurring D RNAs), Damayanti et al. (2002) [43] revealed that deletion of a fragment of ~500 nt in the proximal 5′ and 3′ regions of the BMV 3a ORF (RNA3) supresses encapsidation of artificial D RNAs and affects competition with RNA3 in the artificial D RNAs’ amplification and encapsidation. Such elements essential for DI RNA replication were also mapped for CymRSV [30]. Research performed with artificial D RNAs of TMV led to the conclusion that the replication signals for HV and D RNA may differ [79].

It was confirmed that CymRSV D RNAs are *trans*-regulated by the viral p22 and p92 proteins (replication regulation) [80], whereas CMV D RNAs are *trans*-regulated by the 3a (MP, cell-to-cell movement) [81] and 3b (CP, symptoms on infected plant in the presence of D RNAs) viral proteins [82].

## 5. Effect of D RNAs

### 5.1. Interference with Virus Accumulation

D RNAs that interfere with replication and accumulation of the HV are termed defective interfering RNAs (DI RNAs). Although “absence of evidence is not evidence of absence”, the presence of non-interfering D RNAs appears to have few implications for virus infection and evolution. It has been shown that suppression of HV accumulation is quite frequent for many DI RNA species [27,66,83]; however, interference does not always correlate with high DI RNA accumulation [57]. The impact of DI RNAs on HV RNA replication was investigated with two TBRV isolates originating from different hosts (greenhouse tomato and lettuce). The accumulation of HV was analysed in the presence and absence of DI RNAs in the following plants: tomato, lettuce, quinoa, and tobacco. Results confirmed the hypothesis that the extent of DI RNA interference with the HV depends on virus isolate and host plant. The most spectacular effect was observed for quinoa, where the average reduction of TBRV accumulation was 26% [64].

The interference with HV replication and accumulation by DI RNAs can be explained by competition with the HV for viral and host resources, the mechanism of posttranscriptional gene silencing (PTGS), and modification of the function of viral factors [52,70]. Jones et al. (1990) [84] inoculated *N. benthamiana* protoplasts with different DI RNA: HV ratios. TBSV HV accumulation was reduced by 50% and 65% when plants were infected with 1:4 and 1:1 DI RNA:HV ratios, respectively. The authors claimed that the increased accumulation of DI RNAs and significant reduction of HV accumulation indicated that inhibition of HV occurred as a result of direct competition for viral resources, and not due to HV RNA degradation. Moreover, it was suggested that for TBSV-derived DI RNAs, the interference is mediated by the down-regulation of TBSV’s p19 RNA silencing suppressor (RSS); coinfection of HV and DI RNAs resulted in decreased accumulation of this protein and its sub-genomic RNA [76].

PTGS is a common sequence-specific RNA degradation process used by plants as an antiviral strategy [85]. During the process, double-stranded (ds) RNAs are converted into 21–25 nt RNA fragments (siRNA), and subsequently used to direct ribonucleases to target cognate RNA [86]. For plant viruses, the resulting decreases in the amount of target RNA may lead to the attenuation of infection symptoms. This effect was noticed in plants infected with a mixture of HV and DI RNAs [64,87]**,** suggesting the role of PTGS as a possible mechanism of DI RNA-induced interference. It was confirmed that in CymRSV-infected *N. benthamiana* plants, accumulation of the virus triggered PTGS, which resulted in generation of siRNAs corresponding to the viral genome. The three CymRSV-derived DI RNAs used in the study were targeted differently by helper virus–induced PTGS: the larger precursor form of DI RNAs (679 nt) was targeted successfully, whereas the shorter (mature) form was not. The higher suppression of the longer DI RNA by PTGS resulted from the presence of specific sequences/structures rather than the length of DI RNA. Moreover, the efficient generation of siRNAs from shorter DI RNAs was confirmed [87]. According to these results, for tombusviruses, the model of PTGS-mediated DI RNA evolution (by selective accumulation of DI RNAs without sequences that are targeted by PTGS) and symptom attenuation (as an effect of DI RNA–induced PTGS) was proposed [87].

### 5.2. Attenuation of Infection Symptoms

The attenuation of infection symptoms in the presence of DI RNAs has been confirmed for many virus species. The mechanism seems to be quite common and is reliant on lower levels of HV replication due to competition for replication factors or RNA-mediated enhancement of host resistance [88]. Previous studies showed that the attenuation of infection symptoms was not unique for all DI RNAs accompanying virus of the same species, and differences occurred both in intensity of symptom attenuation and its development over time as different virus isolates were passaged. For example, Knorr et al. [25] showed that when TBSV isolates were serially passaged in tobacco plants, symptom attenuation was observed for some lineages as early as the third passage and as late as the 11th passage. Attenuation of symptoms can be strictly related with the appearance of DI RNAs; however, in some cases, a decrease in symptoms may be observed before DI RNAs can be detected. This observation emphasizes that the underlying processes that lead to symptom development are complex, depending on both properties of individual DI RNAs and the plant’s condition. Moreover, many other factors, including but not limited to interactions between the HV and host, could cause attenuation of symptoms. Symptom attenuation also appears to be host-dependent. In the case of CymRSV, the presence of DI RNAs in the virus inoculum, or introduction of the particles by transgenic nicotiana plants, prevents the appearance of the typical lethal necrotic symptoms of CymRSV infection [29]. Based on the research performed with genomic and DI RNAs of CymRSV, ClRV, and TBSV, Havelda et al. (1998) suggested that DI RNA-mediated enhancement of infection symptoms depends on the ability of the DI RNAs to prevent a direct or indirect interaction between p19 and p33 [89]. P19 and p33 were identified as viral symptom determinants responsible for necrotic symptoms on tombusvirus-infected *N. benthamiana* plants [90].

Interesting and unexpected observations were made for the DI RNA associated with TCV. Although the presence of DI RNAs reduced HV accumulation, symptoms were more severe in the presence of the DI RNAs [18].

### 5.3. Other Effects of D RNAs

The impact of D RNAs on HV, host or vectors seems to be a multi-level process. It was shown that the presence of DI RNAs may affect seed transmission of the HV. Pospieszny et al. (2020) [91] showed that TBRV DI RNAs can be vertically transmitted through seeds. In the experiment, quinoa plants were infected with TBRV isolate (originated from tomato) with and without DI RNA. The plants were grown until seeds could be collected, and development of disease symptoms was observed. Five to six weeks after sowing of the collected seeds, plants were tested for the presence of the TBRV using double antibody sandwich enzyme-linked immunosorbent assay (DAS-ELISA). Overall, over 4000 plants were tested, and it was confirmed that the presence of DI RNAs together with HV at initial infection made the seed transmission of the HV 45% more efficient than in the case of infection without DI RNAs. These results challenge the established framework for considering DI RNAs, as it is possible to have a negative effect on the within-host fitness of the HV (interference with replication) while bolstering between-host fitness (vertical transmission). Moreover, quinoa plants from the second generation were verified for the presence of DI RNAs. The obtained results suggested that DI RNAs are transferred (through the seeds) from generation to generation.

Research performed with transgenic nicotiana plants expressing CymRSV satRNA sequences showed that DI RNA accumulation is supressed by satRNAs. These transgenic plants were still susceptible to infection by CymRSV HV, but the presence of the satRNA did suppress the DI RNAs and also blocked its attenuation of disease symptoms [92].

## 6. Diversity and Evolution of D RNAs

Evolution of the D RNAs seems to proceed similarly to that of the HV virus, as both share the same replication system based on the viral RNA polymerase. However, as D RNAs have to compete with the HV for the viral RdRp, they are continuously under strong selection pressure and can evolve faster than the HV [55]. The faster rate of evolution of D RNAs in comparison with HV can be attributed to higher genetic plasticity of D RNA genomes, resulting from reduced purifying selection pressure eliminating non-functional viral sequences [63]. D RNAs might also undergo mutation and recombination resulting in insertion, deletion or sequence rearrangements. Recombination was widely studied in terms of the generation of primary D RNAs (derived directly from HV genomes in a single mutational event) as well as their descendants—forms of the D RNAs that have been further shortened. Several studies indicated that primary D RNAs evolve to shortened forms upon serial passages. Progressive deletions seem to be the mechanism generating D RNAs of many viruses, e.g., tombusviruses. It was shown that CymRSV is able to induce different types of DI RNAs during passaging in *N. clevandii*. The largest DI RNA sequence (DI-13) was 679 nt in length and consisted of three fragments derived from the HV genome. Sequence analysis of the smaller DI RNAs (from 673 to 404 nt in length) indicated that they are composed of the same three fragments, with a continuing reduction in size through further deletions inside particular blocks of DI-13 (referred to as A, B and C) [28]. Junction sites in DI RNA sequences are certainly not random. These sites were located mostly within blocks A and C in the case of CymRSV [28]. Research performed by Havelda et al. 1997 [93] suggested that in the case of CymRSV DI RNAs, the generation of a shorter variant of particle was associated with intramolecular secondary structures driving the mechanism of deletion [94]. Shorter D RNAs seem to be favoured, as their accumulation is often higher than that of longer variants [66,93]; however, their accumulation is not necessarily dependent on the DI RNA length. For example, shortened forms of TBSV-derived DI RNAs indicated poor targeting by PTGS due to the lack of PTGS target-specific sequences/structures [63] (see Section 5.1). The role of rearrangements and/or recombination events in DI RNA evolution was confirmed for CNV. Passaging of this virus resulted initially in the formation of DI RNAs, but upon further passaging, larger DI RNAs were found [27]. The newly obtained variants contained repeats of three regions found in shorter DI RNA variants, leading the authors to suggest that these larger variants arose as a result of rearrangements between two DI RNAs, rather than being directly derived from the HV genome. The role of mutations as a force driving D RNA evolution also seems to be prevalent. Numerous studies confirmed that D RNAs, despite the fact they are originated from HV genome, indicated single nucleotide changes in their sequences in comparison to genomic RNA. The comparison showed that the identity between the corresponding regions of D RNAs of TBRV and its HV sequence ranged from 98% to 99.5% [49,50].

## 7. D RNA Detection Tools

In classic work on D RNAs, the most popular assay used for their detection was a Northern blot combined with separation and visualisation of D RNAs on sucrose gradients. In later studies, D RNAs were separated on low melting agarose gels, purified using methods of RNA extraction, and then amplified with different variants of a polymerase chain reaction (PCR).

A game changer in virus discovery, identification and sequence analysis has been high-throughput sequencing (HTS) [95]. HTS is a rapidly developing technique that allows for the massively parallel sequencing of millions or even billions of nucleotides in a single sequencing run [96]. It is widely used for viral metagenomics studies to identify known viruses and discover novel species, even in samples with the absence of disease symptoms [95]. This approach was successfully applied for detection of viral infection in many agricultural crops, as well as weeds and wild plants [95,96,97,98]. HTS, unlike to previously used diagnostic methods, gives a more complete perspective on the virome and provides insights into the virus population structure, ecology or evolution [99].

The identification of D RNAs from HTS data can be challenging; D RNAs typically occur together with HV and have high sequence identity with HV, often not containing any unique sequences. In recent years, HTS has been successfully adapted to D RNAs detection. To date, different tools for identification of different D RNA types in HTS outputs have been published, for example, ViReMa (as the first specific tool) [100], DI-Tector [101], DVG-profiler [102] and newly published DVGfinder [103]. The presented tools re-examine unmapped or mapped reads with mismatches and insertions, which potentially include D RNA sequences. Current challenges in D RNA identification from HTS data include improving the sensitivity and precision of the used algorithms, reduction of false positive D RNAs detected, quantification of the relative abundance of given D RNAs and HV, and normalization between samples and sequencing runs [10]. It is worth mentioning that the presented tools do not provide any functional information about the detected D RNA candidates in the replication process.

## 8. Concluding Remarks

Infection by RNA viruses can be accompanied by subviral particles such as D RNAs, DI RNAs and satellite RNAs. These particles are relatively short, non-infectious entities, and their replication, encapsidation and spread depend on the HV. DI RNAs are particularly interesting due to their ability to interfere with virus replication and, therefore, modulation of symptoms on infected plants. Previous research suggested that D RNAs are not abundant in plant viral infections in natural ecosystems. This conclusion could have resulted from both a low concentration of D RNAs in natural infection and/or lack of effective tools for their detection. HTS and dedicated bioinformatic tools for D RNA detection will improve the identification of D RNAs in infected plants in vivo, while also providing stronger evidence of their absence. Despite the recent progress in studies of D RNAs of plant viruses, many questions about D RNAs remain unanswered. First, are D RNAs just a result of errors occurring during virus replication? Or could plants have evolved features that promote their occurrence to weaken the virus? Second, do DI RNAs directly interfere with the replication of HV, or is this process more complex and, for example, mediated by the host? More detailed studies to look for the subtle impacts of D RNAs on infection, transmission or evolution would also be of great interest. Third, is D RNA replication host-specific? Although the de novo occurrence of D RNAs depends on the host, to what extent can existing D RNAs be maintained in different hosts? The knowledge regarding DI RNA formation and impact on HV replication is still limited, whereas the detailed analysis of those mechanisms might be a step toward new, innovative tools to protect plants against viruses. DI RNA particles represent a major controlling element of virus replication. The more we learn about viral pathogenesis and the interaction and competition between DI RNAs and the HV, the more we can focus on our research to dissect DI RNA-mediated attenuation of infection.

## Figures and Tables

**Figure 1 viruses-14-02814-f001:**
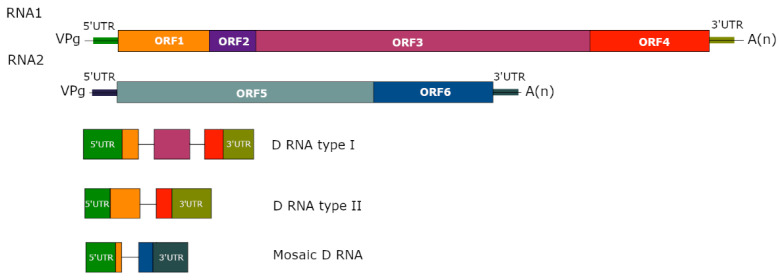
The general representation of different types of D RNAs associated with RNA plant viruses. The colours of different D RNA fragments represent corresponding fragments in HV. The black lines represent the recombination breakpoints.

**Figure 2 viruses-14-02814-f002:**
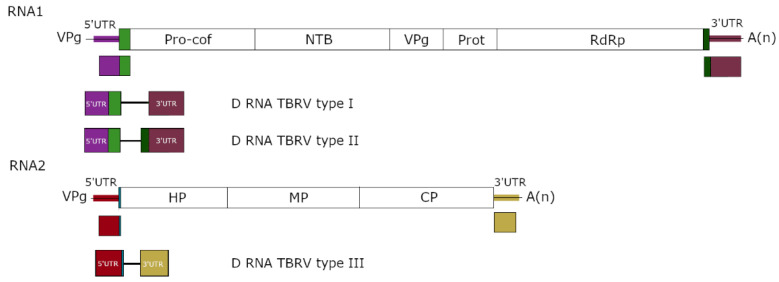
Structure of different TBRV D RNA types. The coloured fragments in the D RNAs correspond to the regions in HV. The black lines represent the recombination breakpoints.

**Table 1 viruses-14-02814-t001:** Different species of D/DI RNAs associated with plant viruses.

Genus	Virus	Origin/Mutation	Characteristics	References
*Tombusvirus*	tomato bushy stunt virus (TBSV)	serial passaging/single round of infection/multiple deletions	attenuation of infection symptoms/reduced virus accumulation	[25,26,27]
	cymbidium ringspot virus (CymRSV)	serial passaging/multiple deletions	attenuation of infection symptoms/reduced virus accumulation	[17,28,29,30]
	cucumber necrosis virus (CNV)	serial passaging/	attenuation of infection symptoms/reduced virus accumulation	[19,27]
	turnip crinkle virus (TCV)	serial passaging/	exacerbation of infection symptoms	[18]
*Orthotospovirus*	tomato spotted wilt virus (TSWV)	serial passaging/single deletion	modulate virulence	[31,32]
*Closterovirus*	citrus tristeza virus (CTV)	serial passaging/single deletion	no interference observed	[33,34,35,36]
*Tobravirus*	tobacco rattle virus (TRV)	serial passaging/deletions and/or rearrangements—mosaic of RNA1 and RNA2 of the HV	reduced virus accumulation	[37,38]
*Potexvirus*	clover yellow mosaic virus (ClMV)	serial passaging/single deletion	no interference observed	[39,40]
	casava common mosaic virus (CsCMV)	serial passaging/single deletion	no interference observed	[41]
*Bromovirus*	brome mosaic virus (BMV)	prolonged single round of infection	-	[42,43]
	broad bean mottle virus (BBMV)	prolonged single round of infection/single deletion	exacerbation of infection symptoms	[44,45]
*Cucumovirus*	cucumber mosaic virus (CMV)	serial passaging/single deletion	-	[20,46]
*Crinivirus*	lettuce infectious yellows virus (LlYV)	single round of infection	-	[47,48]
*Nepovirus*	tomato black ring virus (TBRV)	serial passaging/single deletion	attenuation or enhancement of infection symptoms/suppression of virus accumulation	[49,50,51]
*Pomovirus*	potato mop-top virus (PMTV)	serial passaging/single deletion	attenuation of infection symptoms	[52,53]

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
