# Peer review of "Defective RNA Particles of Plant Viruses—Origin, Structure and Role in Pathogenesis"

_viruses, 2022, doi:10.3390/v14122814_

Round 1

Reviewer 1 Report

This review paper summarize the anvance of defective RNA particles of plant viruses, and give outlook about some unanswered aspects about defective RNAs. The review is comprehensive and may give some guidance for those researcher studing the genome of plant viruses. 

Some minor grammar points:

Line 17: plant crucial role

Line 19: often are

Line 48 [8]however,

Line 60 this results

Line 89 for many years hypotheses

Reviewer 2 Report

The review deals with the existence of defective RNAs in plant viruses. The authors make a clear distinction throughout the MS between defective and defective-interfering RNAs and provide appropriate references and examples.

All the sections are properly located, which takes the reader in a sequential manner to understand the main focus of the article.

I would only recommend to include, as part of the discussion, a few cases where circular and 5' to 3' concatenated RNA segments have been reported. The authors may want to check the following work:

Blackcurrant Leaf Chlorosis Associated Virus: Evidence of the Presence of Circular RNA during Infections by Delano et al. 2018

and

A raspberry bushy dwarf virus isolate from Ecuadorean Rubus glaucus contains an additional RNA that is a rearrangement of RNA-2 by Quito-Avila et al.  2014

Reviewer 3 Report

The review manuscript of Daria et al., well summarized the origin, structure, and roles of plant viruses-derived defective RNA, however, there are several problems including but not limited those suggestions listed below, that will require the authors attention and revise before publishing in Viruses journal.

Line 17: role change to roles

Line 46: was change to were

Line 48: add “,” before “however”

Line 52: has led to change to have led to

Line 54: which has been confirmed for representatives of each family in vitro. Please add references

Line 60: this results makes them change to these results make them

Line 66: and no molecular confirmation of genomic mutations was obtained, this sentence is unclear, change to: but their genomic mutations had not been confirmed in molecular?

Line 74: For many of these viruses artificial D RNAs were constructed to, change to For many of these viruses, their artificial D RNAs were constructed to

Line 78: “D RNAs associated with plant viruses described so far differ even within species”, it is hard to read, change to “The plant viruses-associated D RNAs is highly differed even within species, including nucleotide structure, host dependence or effect on HVs etc.” ?

Line 85: During D RNAs generation two different stages can be distinguished: change to “During D RNAs generation, two different stages can be distinguished: or Two different stages can be distinguished during D RNAs generation:

Line 89: “For many years hypotheses concerning D RNA formation were widely investigated and” add “,” behind the years, or change to “Hypotheses concerning D RNA formation were widely investigated for many years”

Line 107: for different species of virus and different isolates, change to “for different virus species and isolates

Line 109: 3rd and 11th change to 3rd and 11th

Line 111: could purge “DIs”?, is DI RNAs? or D RNAs ?

Line 126: The most probable mechanism of D RNA formation is considered hopping or template switching of viral RdRp, change to “Hopping or template switching of viral RdRp is considered to be the most probable mechanism of D RNA formation”

Line 127: D RNAs are derived mainly from, change to D RNAs are mainly derived from

Line 131: what is “ copy choice mechanism”? Needs explanation

Line 148: nearby the genomic sequences of

Line 171: wt virus? Change to wild type virus

Line 180: Figure 1 is unclear, needs redraw: 1. Enlargement of these three types of D RNAs images. 2. Add annotation of UTR to distinguish from ORFs. The same to Figure 2

Line 236: have change to has

Line 240: in the presence to or in the presence of ?

Line 245: what does “Although lack of evidence is not evidence of lack” means?

Line 258: need add ”,” before and

Line 266: accumulation of ..

Line:276: accumulation of the virus triggered PTGS and resulted in 275 accumulation of siRNA corresponding to the viral genome. change to : the accumulation of the virus triggered PTGS, which resulted in generation of siRNA corresponding to the viral genome ?

Line 304: prevents the appearance of the lethal necrotic symptoms typical of CymRSV infection change to “prevents the appearance of the typical lethal necrotic symptoms of CymRSV infection”

Line 321 to 323: unclear, please re-write

Line 313: Diversity and evolution of D “RNA”, change to ”RNAs”

Line 352: Junction sites in DI RNA sequences certainly are not random, and in case of CymRSV were located mostly within blocks A and C. change to “Junction sites in DI RNA sequences are certainly not random which were located mostly within blocks A and C in the case of CymRSV”?

Line 356: higher than for,change to “higher than”? need confirmation

Line 357: add ”,” before however

Line 358: TBSV derived DI RNAs change to TBSV-derived DI RNAs

Line 360: DI RNA, please add s in RNA

LINE 362: was found

Line 364: DI RNAs

Line 369 its HV sequence

Line 369: change to : showed that the identity between the corresponding regions of D RNAs of TBRV and its HV sequence ranged from 98% to 99,5 % ?

Line 384: provides

Line 387: have very similar sequences to HVs? have high sequence identity with HVs

Line 393: D RNAs

Line 394: quantification of

Line 404: Previous researches
